# Colchicine reduces lung injury in experimental acute respiratory distress syndrome

**Jocelyn Dupuis**[1,2‡]*, **Martin G. Sirois**[1,3], **Eric Rhéaume**[1,2], **Quang T. Nguyen**[1], **Marie-Élaine Clavet-Lanthier**[1], **Genevieve Brand**[1], **Teodora Mihalache-Avram**[1], **Gabriel Théberge-Julien**[1], **Daniel Charpentier**[1], **David Rhainds**[1], **Paul-Eduard Neagoe**[1], **Jean-Claude Tardif**[1,2‡]*

1 Montreal Heart Institute Research Center, Montreal, Quebec, Canada, 2 Department of Medicine, Faculty of Medicine, Université de Montréal, Montreal, Quebec, Canada, 3 Department of Pharmacology and Physiology, Faculty of Medicine, Université de Montréal, Montreal, Quebec, Canada

‡ JD and J-CT are joint senior authors.
* dupuisj@icloud.com (JD); jean-claude.tardif@icm-mhi.org (JCT)

**Data Availability Statement:** All relevant data are within the manuscript and its Supporting Information files. All individual experimental points are reported.

## Abstract

The acute respiratory distress syndrome (ARDS) is characterized by intense dysregulated inflammation leading to acute lung injury (ALI) and respiratory failure. There are no effective pharmacologic therapies for ARDS. Colchicine is a low-cost, widely available drug, effective in the treatment of inflammatory conditions. We studied the effects of colchicine pre-treatment on oleic acid-induced ARDS in rats. Rats were treated with colchicine (1 mg/kg) or placebo for three days prior to intravenous oleic acid-induced ALI (150 mg/kg). Four hours later they were studied and compared to a sham group. Colchicine reduced the area of histological lung injury by 61%, reduced lung edema, and markedly improved oxygenation by increasing $PaO_2/FiO_2$ from 66 ± 13 mmHg (mean ± SEM) to 246 ± 45 mmHg compared to 380 ± 18 mmHg in sham animals. Colchicine also reduced $PaCO_2$ and respiratory acidosis. Lung neutrophil recruitment, assessed by myeloperoxidase immunostaining, was greatly increased after injury from 1.16 ± 0.19% to 8.86 ± 0.66% and significantly reduced by colchicine to 5.95 ± 1.13%. Increased lung NETosis was also reduced by therapy. Circulating leukocytosis after ALI was not reduced by colchicine therapy, but neutrophils reactivity and CD4 and CD8 cell surface expression on lymphocyte populations were restored. Colchicine reduces ALI and respiratory failure in experimental ARDS in relation with reduced lung neutrophil recruitment and reduced circulating leukocyte activation. This study supports the clinical development of colchicine for the prevention of ARDS in conditions causing ALI.

## Introduction

The acute respiratory distress syndrome (ARDS) results from direct or indirect acute lung injury (ALI) leading to intense inflammation with alveolar edema producing respiratory failure. ARDS accounts for 10% of intensive care units admissions and for 24% of mechanically ventilated patients [1, 2]. In the United States alone, it affects approximately 200 000 patients per year with a high mortality rate ranging from 35% to 46%, higher mortality being associated with greater initial ALI [1]. Furthermore, survivors of ARDS suffer from significant long-term

**Funding:** This study was funded by the Government of Quebec and the Canada Research Chair in translational and personalized medicine held by J-CT. The funders had no role in study design, data collection and analysis, decision to publish, or preparation of the manuscript.

**Competing interests:** I have read the journal's policy and the authors of this manuscript have no competing interests to declare.

morbidity affecting quality of life with physical, neuropsychiatric and cognitive impairments [1]. Besides mechanical ventilatory support, there are no treatment options currently available and all trials with pharmacologic agents have shown neutral or even deleterious effects [3]. ARDS represents the major complication of SARS-CoV-2 infection and in the context of the current COVID-19 pandemic, there is an urgent need for effective therapies that could reduce intensive care unit admissions, mechanical ventilation and death rate.

The early pathologic phase of ARDS, termed the "exudative" phase, is characterized by high-permeability alveolar edema with intense dysregulated inflammation [2, 4]. In this early phase following ALI, histological study reveals predominant neutrophilic alveolitis: activated polymorphonuclear neutrophils recruit to lung tissue in concert with monocytes and alveolar macrophages [5]. It is recognized that neutrophils play a major role in lung tissue injury by their rapid recruitment and their release of proteases (myeloperoxidase (MPO), elastase, matrix metalloproteases (MMP)), free oxygen radicals and neutrophils extracellular traps (NETs) [6, 7]. Initial neutrophils recruitment results from the release of pathogen-associated molecular patterns (PAMPs) and damage-associated molecular patterns (DAMPs) activating the inflammasome [6]. PAMPs comprise lipopolysaccharide (LPS), lipoteichoic acid, DNA, RNA and foreign proteins such as formylated peptides released by infectious agents and recognized by immune receptors such as Toll-like receptors. DAMPs are released after tissue injury and include high mobility group box 1 (HMGB1), heat shock proteins, hyaluronan and mitochondrial-derived factors. While neutrophils play a critical role in ALI, macrophages and monocytes orchestrate resolution of inflammation and tissue repair [8] whereas CD4 regulatory T lymphocytes may play a central role in the control of neutrophil recruitment in indirect ALI [9].

Colchicine is a widely available low-cost drug proven effective in the treatment and prevention of inflammatory disorders characterized by dysregulated inflammation including gout, pericarditis and familial Mediterranean fever. More recently, colchicine prevented recurrent events after acute coronary syndrome, a disorder also characterized by inflammation with neutrophils and macrophages rich plaques [10]. Colchicine inhibits tubulin polymerization in leukocytes reducing their adhesion, recruitment and activation. Colchicine abrogated NETs formation in acute coronary syndrome patients after percutaneous coronary intervention [11] and in patients with active Behçet's disease [12]. It also reduces inflammation by interfering with leukocytes inflammasome activation, in particular the NLRP3 complex [13, 14] and therefore may lead to decreased levels of proinflammatory cytokines 1L-1β and IL-6 [15].

The COLCORONA trial is an international randomized double-blind placebo-controlled clinical study evaluating colchicine for the prevention of COVID-19-related complications. In that context, we evaluated whether colchicine pre-treatment could reduce ALI, inflammation and respiratory failure in a well characterized model of ARDS. Rats were pre-treated with colchicine (1mg/kg/day) or placebo for 3 days prior to oleic acid-induced ALI and compared to a control group 4 hours after ALI.

## Materials and methods

The study protocol was approved by the animal research and ethics committees of the Montreal Heart Institute and all experiments conducted in accordance with the Canadian guidelines for the care of laboratory animals.

### Animal model and colchicine administration

Male Wistar rats weighing 250–300 g were purchased from Charles River, St. Constant, Québec. They were divided in three groups: Sham + placebo (n = 8), oleic acid + placebo (n = 8)

and oleic acid + colchicine (n = 8). Oleic acid (150 mg/kg in 0.3 cc) was prepared daily as a mixture with 0.1% bovine serum albumin (BSA) dissolved in distilled water, stored and protected from light at room temperature. Colchicine (1 mg/kg) or distilled water (placebo) was administered by daily gavage (in a volume up to 2 ml/kg) for three days before the induction of the ALI. On the beginning of the fourth day, a last dose of colchicine or placebo was administered and rats immediately injected via a jugular vein cannula with oleic acid or with 0.1% BSA. There was no death in the sham group. There was one death in the oleic acid + placebo group 3 hours after injection of oleic acid and prior to any measurement. There was one surgical death in oleic acid + colchicine treatment by laceration of the jugular vein, prior to injection of oleic acid.

Oleic acid was administered under ketamine/xylazine anesthesia by slow bolus injection over 30 seconds and the catheter was flushed with 0.3 ml of 0.1% BSA before and after injection. Four hours later, the animals were studied. Conscious unrestrained respiratory parameters were obtained by whole body plethysmography (Emka technologies). Rats were then anesthetized with 2.5% isoflurane and 100% oxygen (1L/min) administered with a nose cone/face mask for 5 minutes before terminal exsanguination. Arterial blood was collected through the thoracic aorta into a syringe containing lyophilized heparin for arterial blood gas measurement and into EDTA tubes for complete blood count and flow cytometry. Serum cytokines were assessed with serum clot activator tubes. The left lung was cannulated and perfusion-fixed with 10% buffered formalin for histology and immunohistochemistry. The right superior and middle lobes were used to measure the lung weight and edema. The inferior lobes of the right lung were snap-frozen and stored at -80˚C for gene expression analysis. Pulmonary edema was measured from the ratio of total divided by dry weight of the right lung. Total weight, including water, was measured and lung tissue was dried in an oven at 60˚C for 5 days and reweighted as dry weight.

## Lung histology and immunohistology

All histological and immunohistological procedures were performed by the same person blinded to treatment assignment. Lung tissues were dehydrated by incubating in a series of solutions with an increased ethanol content (70, 95 and 100%), followed by xylene, and embedded in paraffin. The specimens were cut into 6-μm sections, mounted on charged slides, and processed with hematoxylin phloxine saffron (HPS) staining.

Immunohistological procedures were initiated by incubating the slides in citrate antigen retrieval (pH 6.0) and endogenous peroxidase blocking (3% hydrogen peroxide). Sections were then blocked by incubating in PBS containing 10% normal goat serum (same species as secondary antibody) for 60 minutes. Slides were incubated with a rabbit polyclonal anti-myeloperoxidase (MPO, Pa5-16672; ThermoFisher, MA USA) for neutrophil detection, with a rabbit polyclonal anti-histone H3 citrulline R2 + R8 + R17, (Cit-H3, ab5103; ABCAM, Cambridge, United Kingdom) for NETosis, and primary antibodies were omitted for negative controls. After washing, sections were incubated with a biotinylated secondary antibody (Vector Laboratories, Burlingame, CA) for 30 minutes, washed, then incubated with avidin-biotin complex (ABC kit) and visualized using diaminobenzidine substrate (Vector Laboratories) [16–18].

The HPS slides were scanned to get a picture of the whole left lung (Super coolscan 5000; Nikon, Tokyo Japan). Using a brightfield microscope (BX45, Olympus, Richmond Hill, ON, Canada), images were acquired under 200× magnification on the most damaged/altered regions, acquiring 5 fields per slide for the HPS staining and 10 fields per slide for IHC (MPO and Cit-H3) staining. For the HPS staining, the following analyses were performed: 1) thickness of alveolar membranes, 2) percentage (%) of altered lung tissue, 3) injury score of the lungs.

To assess the thickness of alveolar membranes, 20 measurements per field were performed (corresponding to 100 measurements per slide) and were expressed as the mean thickness (μm) of the alveolar membranes. A morphometric analysis has been performed to assess the percentage of altered lung tissue over total lung area (excluding trachea, major bronchi and blood vessels >700 μm diameter). To evaluate lung injury, we used an adapted version of the standardized histology score from the American Thoracic Society Documents [19]. The histology scores (0, 1 or 2), were given for: 1) neutrophils in the alveolar space, 2) neutrophils in the interstitial space, 3) proteinaceous debris, 4) alveolar septal thickening, 5) alveolar hemorrhage, 6) interstitial space/membrane hemorrhage and 7) alveolar necrosis. For each slide the maximal injury score, corresponding to the sum of the score (score 0 to 2) of the 7 parameters x 5 fields per slide, is 70 points (2x7x5).

Neutrophils immunoreactivity in the lungs was performed to assess the presence of neutrophils (MPO immunostaining) and neutrophils undergoing NETosis (Cit-H3). The percent area (%) occupied by neutrophils in the lungs was quantified by color segmentation and represented as the MPO area over total lung tissue area. To assess neutrophils undergoing NETosis, we quantified the intensity of Cit-H3 staining (lumen). All analyses were performed using Image Pro Premier version 3.0 software (Media Cybernetics, Rockville, MD, USA).

### Serum cytokine quantification

Quantitative determination of serum IFN-γ, IL-1β, IL-4, IL-5, IL-6, KC/GRO, IL-10, IL-13, and TNF-α was performed using the electrochemiluminescence-based Meso Scale Discovery (MSD) platform (Rockville, Maryland, USA). Proteins levels were measured in a multiplex assay using the V-PLEX Proinflammatory Panel 2 Rat kit (MSD). Samples were diluted 1:5 in proprietary buffer (MSD) and measured. Data were acquired using a MESO QuickPlex SQ 120 plate reader (MSD) and protein concentrations were determined using the MSD Discovery Workbench 4.0 analysis software. All values reported are between the lower and the upper limits of quantification of the kit. Quantitative determination of serum Cit-H3 was measured by a commercially available ELISA kit (Cayman Chemical, 501620, Ann Arbor, MI, USA).

### Flow cytometry

*In vitro* **LPS challenge of blood leukocytes.** Whole blood samples collected on EDTA-coated tubes were incubated or not with LPS (500 ng/mL) for 30 minutes at 37˚C. The samples were then stained using mouse anti-rat CD45 Alexa Fluor 700 (clone OX-1, conjugated to Alexa Fluor 700, Biolegend), mouse anti-rat CD11b antibody (clone WT.5, conjugated to V450, BD Horizon), and 7AAD (Biolegend) for 30 minutes at 4˚C. FACS lysing solution (BD Pharm Lyse, BD Biosciences) was added for 15 minutes at room temperature and samples were stored at 4˚C until analysis by flow cytometry (LSRII, BD Biosciences). Data were analyzed on Diva software version 8.0.1 (BD Biosciences).

**Leukocyte immunophenotyping.** Using reverse pipetting, 100 μl of whole blood collected on EDTA-coated tubes were incubated with anti-CD32 to prevent FC-mediated non-specific binding according to the manufacturer's instructions. The panel of antibodies as well as the gating strategy were inspired by Barnett-Vanes and al. [20] and detailed in the S1 Table. Briefly, a panel of 11 antibodies was used to identify the neutrophils (CD45+/ SSChi / His48+), the monocytes (CD45+/ SSClo/ His48hi or lo/CD43hi or lo), the B lymphocytes (CD45+/ SSClo /CD45R-B220+), the T lymphocytes (CD45+/ SSClo / CD3+/CD4+ or CD8+), and the natural killer cells (CD45+/SSClo /CD161a+). The whole blood samples were incubated with the antibodies for 20 minutes at room temperature and the BD Pharm Lyse solution was then added for 10 minutes to lyse the erythrocytes. Samples were stored at 4˚C until analysis by

flow cytometry (LSRII BD Biosciences). Accucount (Spherotech) were added to establish a cell count per μl of blood. Data were analyzed on Diva software version 8.0.1 (BD Biosciences).

## RNA extraction and quantification of mRNA expression by RT-qPCR analysis

About 30 mg of rat lung samples were homogenized in a TissueLyser II (Qiagen, Germany) in 700 μl of RLT buffer from the RNeasy Mini RNA extraction kit (Qiagen, Germany) supplemented with the anti-foam DX Reagent. Total RNA was- extracted according to the kit's instructions. RNA integrity and quantity were assessed using a 2100 Bioanalyzer Instrument (Agilent, Santa Clara, CA). 500 ng of RNA were used for cDNA synthesis using the High-Capacity cDNA Reverse Transcription Kit (Applied Biosystems, Foster City, CA). For RT-qPCR, reactions were performed in duplicates using the SsoAdvanced Universal SYBR Green Supermix (Bio-Rad, Hercules, CA), 2.5 ng of cDNA, and a final concentration of 100 nM of each primer. The cycling protocol started with a denaturation step at 95˚C for 5 min, followed by 40 cycles of denaturation at 95˚C for 15 s, and primer annealing and extension at 57˚C for 30 sec. Upon completion of the cycling steps, a melting curve protocol was performed from 60˚C to 95˚C and the reaction was stored at 4˚C. Real-time PCR was carried out using a CFX384 Touch Real-Time PCR Detection System (Bio-Rad, Hercules, CA). The geometric mean of housekeeping genes ACTB, GAPDH and XBP1 was used as an internal control to normalize the variability in expression levels which were analyzed using the $2^{-\Delta\Delta CT}$ method. The primer sequences are provided in S2 Table. Heatmap of normalized expression values for gene expression was generated using Morpheus from Broad Institute (https://software.broadinstitute.org/morpheus). To normalize the data, the $\log_2$ of normalized expression values for IL-2, IL-13, CXCR2, E-sel, Arg1, CXCL1, CalCRL, Ramp2 and ATP1B1 were used.

## Statistical analysis

Comparisons between groups were performed by one-way ANOVA followed by Tukey's post-hoc comparisons. Significant differences were considered if $p < 0.05$. All values are presented as mean ± SEM.

## Results

### Effects of colchicine on lung edema and gas exchanges (Fig 1)

Oleic acid induced severe ALI with almost tripling of lung weight and severe edema evidenced by increased wet/dry lung weight ratio. Macroscopically, the lungs were larger and hemorrhagic. There was severe respiratory failure with markedly reduced blood oxygenation. The animals were administered 100% $FiO_2$ before arterial blood gas sampling so that the measured $PaO_2$ is equivalent to the $PaO_2/FiO_2$, a recognized parameter indicative of ARDS severity. Clinical ARDS is recognized when the $PaO_2/FiO_2$ is less than 300 mmHg. Oleic acid caused severe respiratory failure as the $PaO_2$ decreased from 380 ± 18 mmHg to 66 ± 13 mmHg (mean ± SEM) with $CO_2$ retention causing respiratory acidosis. Colchicine therapy reduced pulmonary edema, markedly improved blood oxygenation with a $PaO_2$ of 246 ± 45 mmHg, and reduced respiratory acidosis.

### Effects of colchicine on respiratory parameters, blood cell counts, and biochemistry (Table 1)

Animals with oleic acid injury had increased respiratory rate with lower tidal volume. Both were improved, although non-significantly, with colchicine therapy. Oleic acid injection caused an increase in blood hemoglobin and hematocrit, the latter being statistically

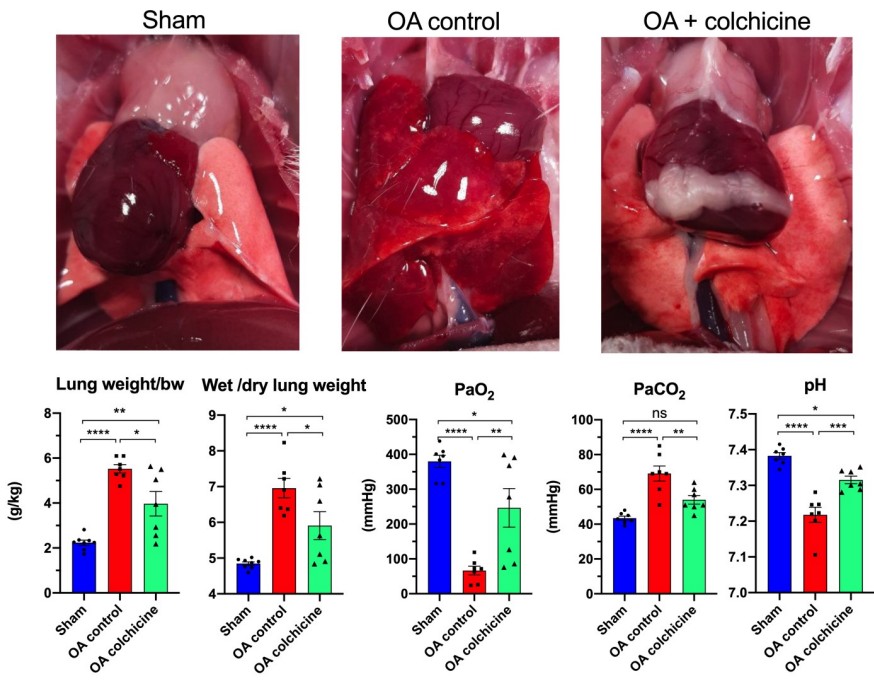

Sham          OA control          OA + colchicine

**** p<0.0001; *** p<0.001, ** p<0.01; * p<0.05

**Fig 1. Effects of colchicine therapy on oleic acid-induced ALI, lung edema and gas exchange.** Values are mean ± SEM. $^{****}$ p$<$0.0001; $^{***}$ p$<$0.001, $^{**}$ p$<$0.01; $^{*}$ p$<$0.05.

significant. Colchicine therapy normalized both hemoglobin and hematocrit, the latter being no longer significantly different from those of the sham animals. Oleic acid increased

**Table 1. Effect of colchicine on respiratory parameters, blood cell counts, and biochemistry.**

|  | Sham | OA control | OA + colchicine |
|---|---|---|---|
| **Breathing rate (bpm)** | 102 ± 6 | 181 ± 10*** | 166 ± 20** |
| **Tidal volume (ml)** | 2.67 ± 0.15 | 1.19 ± 0.05*** | 1.81 ± 0.35* |
| **Peek inspiratory flow (ml/s)** | 16.01 ± 1.23 | 16.43 ± 0.80 | 17.75 ± 2.72 |
| **Peek expiratory flow** | 15.48 ± 1.16 | 11.83 ± 0.68 | 13.19 ± 1.41 |
| **Hemoglobin (g/L)** | 150 ± 2 | 158 ± 3 | 153 ± 2 |
| **Hematocrit (%)** | 45.0 ± 0.6 | 48.7 ± 0.8** | 46.6 ± 0.5 |
| **Leukocytes ($10^9$/L)** | 2.32 ± 0.29 | 3.75 ± 0.49 | 4.90 ± 0.68** |
| **Platelets ($10^9$/L)** | 1029 ± 15 | 844 ± 57* | 986 ± 56 |
| **Neutrophils ($10^9$/L)** | 1.14 ± 0.17 | 2.00 ± 0.35 | 2.71 ± 0.36** |
| **Lymphocytes ($10^9$/L)** | 1.31 ± 0.15 | 1.77 ± 0.18 | 2.19 ± 0.33 |
| **Glucose (mmol/L)** | 16.8 ± 1.3 | 27.0 ± 0.6**** | 21.7 ± 1.6*, †† |
| **Sodium (mmol/L)** | 143.7 ± 0.7 | 138.3 ± 0.9*** | 141.4 ± 0.9† |
| **Calcium (mmol/L)[1]** | 1.38 ± 0.01 | 1.42 ± 0.01 | 1.40 ± 0.01 |
| **Lactate (mmol/L)** | 2.31 ± 0.26 | 1.90 ± 0.37 | 2.4 ± 0.53 |
| **Bicarbonate (mmol/L)** | 26.0 ± 0.7 | 28.1 ± 1.1 | 27.6 ± 0.8 |

*, **, ***, **** is for p $<$0.05, 0.01, 0.001, 0.0001 versus Sham respectively.

†,†† is for p $<$0.05, 0.01 versus OA+Placebo respectively. All values are mean ± SEM.

[1]Calcium corrected for pH.

leukocyte count, the increase being statistically significant in the colchicine treated animals compared to the sham group. This increase was mostly caused by an elevation in neutrophils, also only significant in the colchicine treated animals. There was a mild, non-significant increase in lymphocytes after oleic acid, with no effect of colchicine therapy. Platelet count was significantly reduced by oleic acid injury, and partly normalized after colchicine therapy, being no-longer different from the sham group. Oleic acid caused a marked elevation of plasma glucose that was significantly reduced by colchicine therapy. Inversely, plasma natremia was reduced by oleic acid and improved by colchicine. There was no difference on corrected calcium, lactates and bicarbonates between groups.

## Effects of colchicine on lung injury, neutrophil recruitment, and NETs formation (Fig 2)

All samples were analyzed by a technician blinded to treatment assignment. Oleic acid caused important histological lung injury. Representative examples of the appearance of the complete left lung at low magnification demonstrate evident zones of severe damage, markedly reduced in colchicine-treated animals. The proportion of injured area in the oleic acid ALI group was $24.6 \pm 2.4\%$ with a 61% relative reduction down to $9.6 \pm 3.1\%$ after colchicine treatment. The lung injury score (maximum score of 70) was markedly elevated after oleic acid from $5.1 \pm 0.8$ in sham to $44 \pm 2.8$, and markedly reduced by colchicine to $28.7 \pm 5.0$. Mean alveolar wall thickness more than doubled after oleic acid from $4.6 \pm 0.3$ μm to $10.9 \pm 1.1$ μm and this was also significantly reduced by colchicine to $7.8 \pm 1.1$ μm. Both the lung injury score and alveolar wall thickness were inversely correlated with the severity of ARDS measured from the $PaO_2$. Lung neutrophil recruitment, as assessed by MPO immunostaining, was greatly increased after ALI from $1.16 \pm 0.19\%$ to $8.86 \pm 0.66\%$ and significantly reduced by colchicine therapy to $5.95 \pm 1.13\%$. Lung neutrophils undergoing NETosis were assessed by measuring citrullinated histone H3 (Cit-H3) immunostaining intensity. Treatment with oleic acid increased basal NETosis by 63%, whereas colchicine therapy nearly halved the induction of NETosis, bringing it back to a 35% increase over basal level, a value no longer statistically different from the control group.

## Effects of colchicine on circulating cytokines (Fig 3)

Oleic acid-induced ALI was associated with an increase in mean serum IL-6, TNF-α, and KC/GRO. There was a trend towards reduced IL-6 level with colchicine, and TNF-α and KC/GRO were also non-significantly reduced by therapy. There was no change in any of the other cytokines measured: Cit-H3, INF-γ, IL-10, IL-13, IL-4 and IL-5. IL-1β level was too low to be quantified in 14 of the 20 rats.

## Effects of colchicine on blood leukocyte activation evaluated by flow cytometry (Fig 4)

Leukocyte count on flow cytometry demonstrated an increase in neutrophils that was significant in colchicine treated animals, but there was no change in lymphocytes counts. A leukocyte activation assay was performed on rat blood samples obtained at the time of sacrifice by in vitro LPS stimulation (500 ng/mL) and analysis of the surface expression of CD11b, a member of the β2 integrin family. In sham-treated rats, LPS stimulation resulted in approximately a 2-fold increase of cell surface CD11b mean fluorescence intensity (MFI). Oleic acid exposure tended to decrease the response to LPS in leukocytes and significantly reduced the response in granulocytes, whereas colchicine restored the normal response to LPS stimulation. In

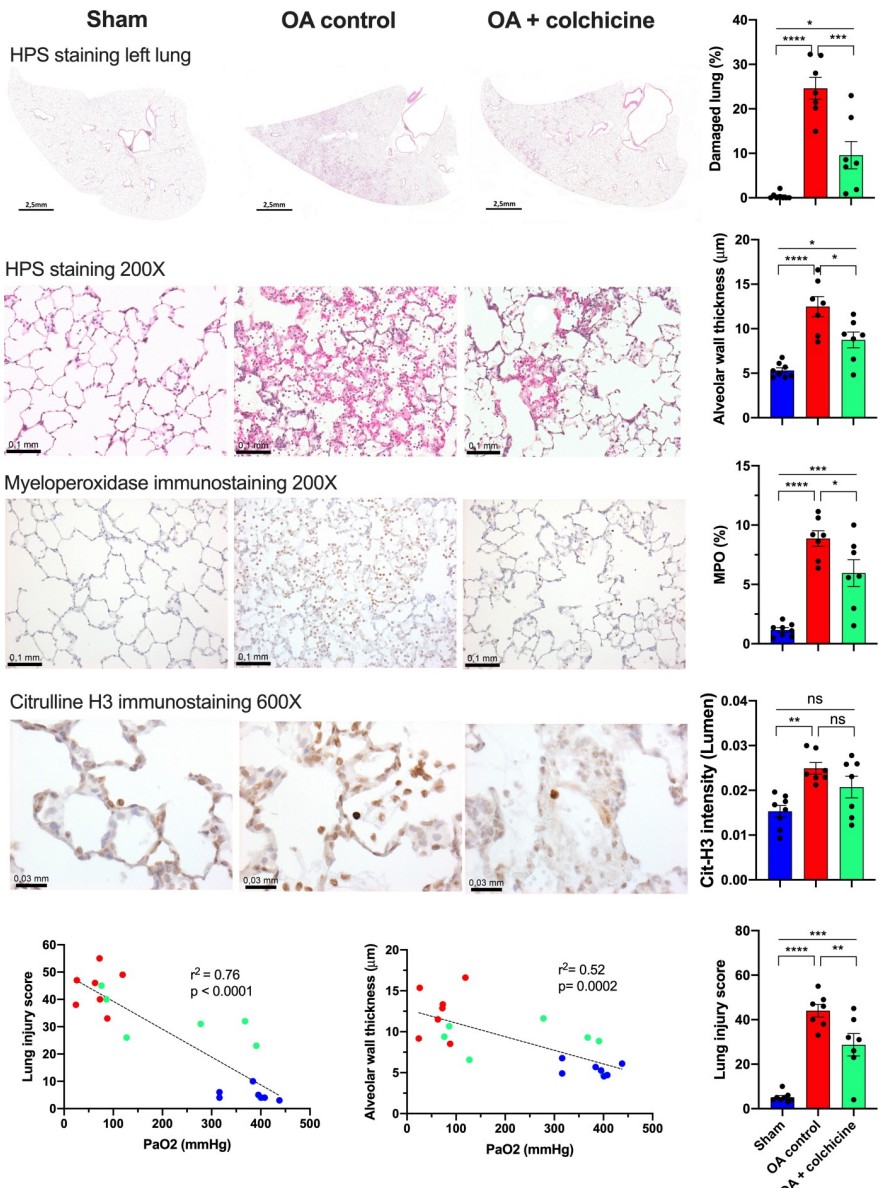

**Fig 2. Effects of colchicine therapy on lung injury, neutrophils recruitment, and NETosis after oleic acid-induced ALI.** Values are mean ± SEM. **** $p<0.0001$; *** $p<0.001$, ** $p<0.01$; * $p<0.05$.

unstimulated blood from oleic acid treated animals, we observed an increase in His48 fluorescence of neutrophils that was normalized by colchicine treatment. Additionally, cell surface CD4 and CD8 on respective lymphocyte sub-populations were increased after oleic acid stimulation, and also normalized by colchicine therapy.

## Effects of colchicine on markers of inflammation and injury in lung tissue (Fig 5)

The mRNA expression of 27 genes involved in inflammatory responses and injury was measured in lung tissue. The results, summarized in a clustered heatmap, clearly demonstrate lung tissue transcriptional regulation of these genes in oleic acid-exposed rats. There was, however, no individual statistically significant effect of colchicine therapy.

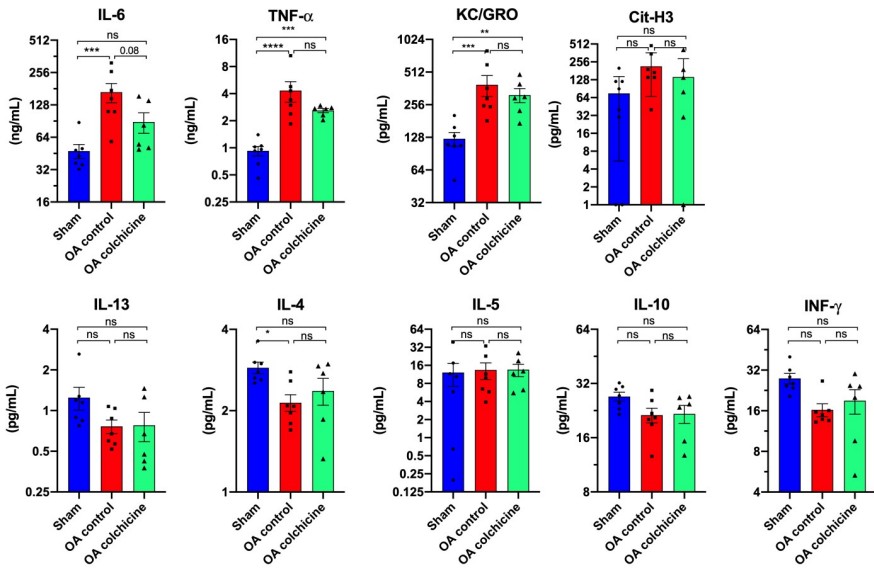

**** p<0.0001; *** p<0.001; ** p<0.01; * p<0.05

**Fig 3. Effects of colchicine therapy on plasma cytokines and NETs 4 hours after oleic acid ALI.** Values are mean ± SEM. **** $p<0.0001$; *** $p<0.001$, ** $p<0.01$; * $p<0.05$.

## Discussion

Acute direct or indirect lung injury causing ARDS represents a serious critical care condition with no effective pharmacologic treatment option [2]. We used the oleic acid model of ARDS to evaluate the effects of colchicine on the acute exudative inflammatory phase of this condition. Colchicine pre-treatment for three days effectively reduced lung edema and markedly improved gas exchanges with increased blood oxygenation and reduced respiratory acidosis. Colchicine reduced ALI evidenced by a reduction in the injured area of 61%, a reduction of the lung injury score in affected regions, and reduced alveolar wall thickness correlating with the improved gas exchange. The benefit of colchicine therapy was related to a reduction in lung recruitment and activation of neutrophils.

### Relevance of this animal model to SARS- CoV-2 respiratory failure

Oleic acid (18:1 n-9), an unsaturated long-chain fatty acid, is the most abundant fatty acid in nature and in the human body. Oleic acid in physiologic concentrations is bound to plasma albumin and non-toxic. Human subjects who develop ARDS have higher plasma levels of C18 unsaturated fatty acids [21]. Injection of high concentrations of oleic acid causes rapid high permeability lung edema with inflammation and impaired gas exchanges and mechanics [22] reproducing human ARDS. Lung injury occurs rapidly after less than 5 minutes [23] with neutrophils infiltration and production of cytokines and chemokines [22]. Acute lung damage caused by oleic acid is characterized by important neutrophils accumulation and activation leading to intense inflammation [22], concordant with the pathophysiology of human ARDS [2]. Besides direct injury through lung embolization, oleic acid stimulates free fatty acid receptors-1 and GPR120 and activates numerous intracellular pathways of inflammation [22].

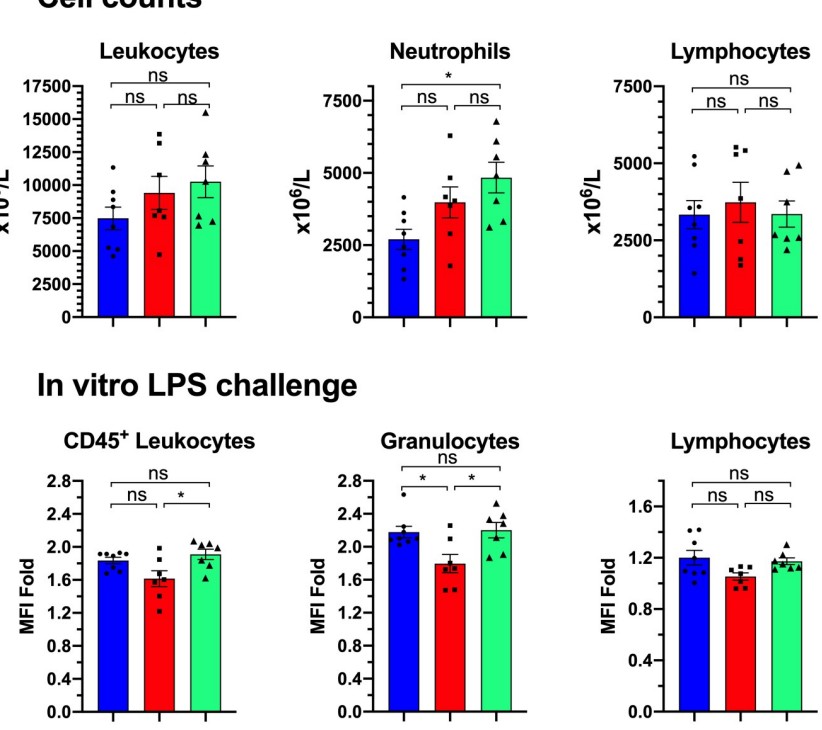

## Cell counts

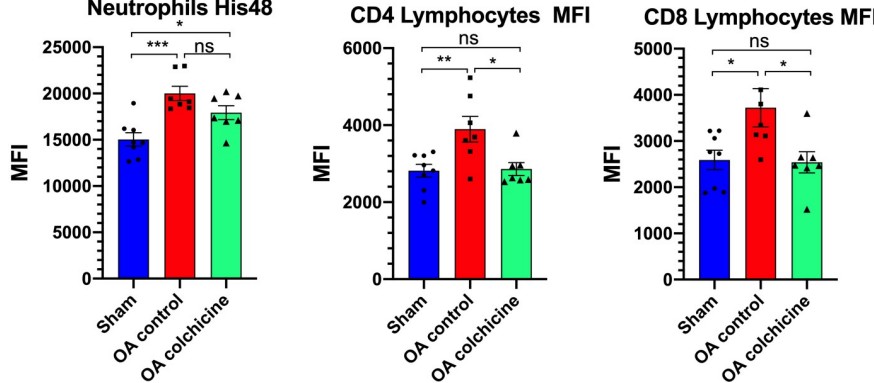

## In vitro LPS challenge

## Cell surface receptors expression

**Fig 4. Whole blood flow cytometry analysis of leukocytes after oleic acid-induced ALI and effects of colchicine therapy.** MFI, mean fluorescence intensity. Values are mean ± SEM. *** $p < 0.001$, ** $p < 0.01$; * $p < 0.05$.

Shared similarities between the oleic acid model of ARDS and coronavirus induced ARDS are the immune dysregulation with intense inflammation associated with lung neutrophil recruitment as well as cytokines elevations. These were well described for the SARS-CoV-1 and MERS-CoV infections [24]. Among the potential inflammatory pathways shared by oleic acid toxicity and coronaviruses is the NLRP3 inflammasome pathway. This cytoplasmic pattern recognition receptor is present in neutrophils and is activated in the lungs after oleic acid injury [25]. It has been shown that activation of NLRP3 by SARS-CoV is a mechanism triggering the inflammatory response to this infection [26]. Although not specifically addressed in this study, colchicine inhibits the NLRP3 inflammasome pathway [14] and this could therefore

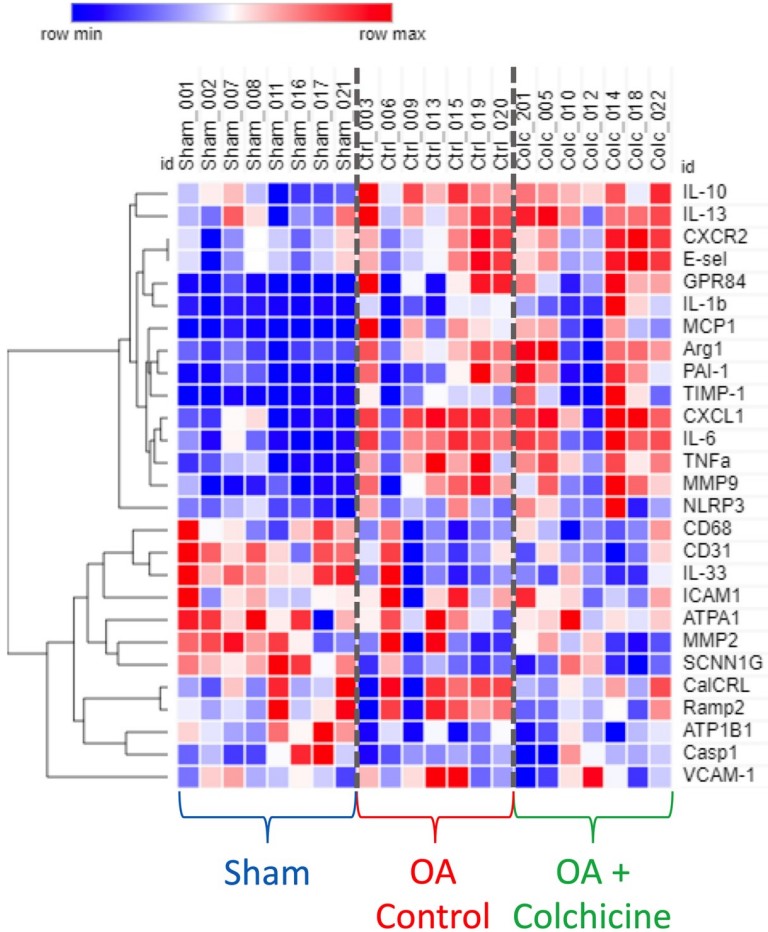

**Fig 5. Clustered heatmap representation of unsupervised clustering of lung tissue mRNA expression of 27 selected markers of inflammation and injury.**

represent a mechanism of benefit in this model. The numerical reduction in plasma concentration of IL-6 with colchicine observed in the current study is in line with an effect on the inflammasome.

## Mechanisms of colchicine benefit

As previously described, oleic acid injection caused circulating leukocytosis predominantly involving neutrophils and important lung recruitment of neutrophils. Colchicine therapy did not reduce circulating neutrophilia, but nevertheless markedly reduced lung recruitment of neutrophils as assessed from MPO immunostaining. Release of neutrophils elastase, myeloperoxidase and NETs contribute to ALI leading to pulmonary edema, alveolar wall thickening, and altered gas exchanges with reduced $PaO_2$ and increased $PaCO_2$. Following the reduction in lung neutrophils recruitment, all of these parameters were improved by colchicine therapy. The lack of effect of colchicine on increased circulating neutrophils suggests that treatment did not interfere with early signaling events responsible for bone marrow release of neutrophils. In addition, it is possible that colchicine caused less neutrophils adhesion to microvasculature and/or transvascular emigration, as observed in lungs, and this might have contributed to the higher neutrophil cell count in circulating blood. Our flow-cytometry analysis on whole blood

from these animals however demonstrates that colchicine modified the biology of oleic acid-treated neutrophils and "normalized" their response to LPS while reducing the expression of the His48 antigen. A previous study in a very inflammatory model of *Pneumocystis* pneumonia demonstrated accumulation of CD11bc[+] and His48[+] cells in the lungs identified as myeloid-derived suppressor cells [27]. Transfer of these cells, which morphologically resemble neutrophils, to control mice caused acute lung damage [27]. Reduction of neutrophil's His48 expression by colchicine treatment may therefore be associated with less "primed" or less damaging neutrophils and provide protection against lung injury. Oleic acid injury was associated with an increase in the plasma cytokines IL-6, TNF-α and KC/GRO (CXCL-1) that was reduced, although non-significantly, by colchicine. Together, these data support that colchicine reduced neutrophils activation with reduced lung recruitment in response to oleic acid injury.

We measured lung tissue mRNA expression of 27 selected markers of lung injury and inflammation. A clustered analysis reveals strong differential activation after oleic acid injury. There was however no statistically significant effect of colchicine on these individual parameters. We believe that this may be in part due to the heterogeneous nature of ALI in the model as evidenced in Fig 2, where damaged zones are interspaced by healthier zones. Analysis of lung tissue homogenates may "dilute" the effect of therapy by including healthier zones into the samples. Selected sampling of injured zones by microdissection may be required to detect treatment effects on these different pathways. Sampling time after injury may also explain the findings as we limited our evaluation at 4 hours and longer duration may be necessary to detect transcriptomic benefits of colchicine treatment. Finally, our sample size may be underpowered to detect a statistically significant difference in these parameters that may have been differently affected in individual animals.

## Variability in the response to colchicine and effect in other ALI models

Although oleic acid caused similar level of injury in animals, we observed individual variability in the benefit of colchicine therapy on lung edema and blood oxygenation. Since Wistar rats are an outbred strain, genetic variability may account for variable benefits of therapy. Also, previous investigations have shown the importance of individual metabolomic profiles of same strain laboratory animals, dependent on environment and microbiota, on the variable response to pharmacologic intervention [28]. In subjects with familial Mediterranean fever, small intestine bacterial overgrowth has been associated with reduced response to colchicine therapy [29].

Other studies have evaluated the effect of colchicine in animal models of ARDS and found similarly beneficial effects. In a mouse model of ALI induced by inhalation of the toxic gas phosgene, colchicine reduced lung neutrophil recruitment, lung injury, and mortality [30]. The benefit was seen even with colchicine administration 30 minutes after phosgene exposure. In yet another different model of ALI caused by hyperoxia in rat pups, colchicine reduced lung injury and lowered TNF-α and IL-1β levels in broncho-alveolar lavage fluids [31]. Together with the current study, these data support the effectiveness of colchicine for the treatment of ALI of various etiologies.

## Limitations and perspective

Our results must be interpreted in the specific context of pre-treatment with colchicine to prevent ARDS and cannot be extrapolated to efficacy for the treatment of florid ongoing ARDS. Although colchicine reduced lung neutrophils recruitment and activation, other potential mechanisms of action of colchicine on the innate immune host response, especially on alveolar macrophages, were not evaluated. These will require further studies.

The results of the current pre-clinical study of ARDS also cannot be extrapolated to human ARDS, but strongly support the conduct of properly sized randomized clinical trials. The

Colchicine Coronavirus SARS-CoV2 Trial (COLCORONA, NCT04322682) is recruiting subjects with SARS-CoV-2 infection to colchicine treatment or placebo to reduce hospitalizations and deaths. Since dysregulated immunity plays a major role in ARDS due to COVID-19 our results strongly support recruitment efforts into this trial.

## Conclusion

Colchicine pre-treatment reduces ALI and lung edema and markedly improves blood oxygenation in the oleic acid model of ARDS by reducing lung neutrophil recruitment and activation. These results strongly support the re-purposing of colchicine, a low cost and widely available drug, in trials of human ARDS.

## Supporting information

**S1 Table. Monoclonal antibodies used for identification of leukocytes sub-populations.** (DOCX)

**S2 Table. Primer sequences used for quantitative real-time PCR of rat target genes.** (DOCX)

## Author Contributions

**Conceptualization:** Jocelyn Dupuis, Martin G. Sirois, Eric Rhéaume.

**Data curation:** Martin G. Sirois, Eric Rhéaume, Quang T. Nguyen, Marie-Élaine Clavet-Lanthier.

**Formal analysis:** Jocelyn Dupuis, Martin G. Sirois, Eric Rhéaume, Quang T. Nguyen, Marie-Élaine Clavet-Lanthier, Genevieve Brand, Teodora Mihalache-Avram, Gabriel Théberge-Julien, Daniel Charpentier, David Rhainds, Paul-Eduard Neagoe.

**Investigation:** Quang T. Nguyen, Genevieve Brand, Teodora Mihalache-Avram, Gabriel Théberge-Julien, Daniel Charpentier, David Rhainds, Paul-Eduard Neagoe, Jean-Claude Tardif.

**Methodology:** Jocelyn Dupuis, Marie-Élaine Clavet-Lanthier, Genevieve Brand, Teodora Mihalache-Avram, Daniel Charpentier, David Rhainds, Paul-Eduard Neagoe.

**Project administration:** Jocelyn Dupuis, Quang T. Nguyen, Jean-Claude Tardif.

**Resources:** Jean-Claude Tardif.

**Software:** Quang T. Nguyen.

**Supervision:** Jocelyn Dupuis, Jean-Claude Tardif.

**Validation:** Jocelyn Dupuis, Eric Rhéaume, Marie-Élaine Clavet-Lanthier.

**Writing – original draft:** Jocelyn Dupuis.

**Writing – review & editing:** Jocelyn Dupuis, Martin G. Sirois, Eric Rhéaume, Jean-Claude Tardif.

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
