## [Decision Letter · Decision Letter 0]

26 Oct 2020

PONE-D-20-28950

Colchicine reduces lung injury in experimental acute respiratory distress syndrome

PLOS ONE

Dear Dr. Dupuis,

Thank you for submitting your manuscript to PLOS ONE. After careful consideration, we feel that it has merit but does not fully meet PLOS ONE’s publication criteria as it currently stands. Therefore, we invite you to submit a revised version of the manuscript that addresses the points raised during the review process.

As you can see from the reviewers comments below, the required changes are relatively minor.  Upon reading the manuscript myself, I also noted a couple of minor issues, which are listed below as well. 

We look forward to receiving your revised manuscript.

Kind regards,

Ruud AW Veldhuizen

Academic Editor

PLOS ONE

Additional Editor Comments:

As the academic editor, I have two additional comments to those provided by the two reviewers.

1) Please include a short discussion regarding the limitation of utilizing **pretreatment** for ARDS.

2) Please include scale bars in the histological images. 

Journal Requirements:

Reviewers' comments:

Reviewer's Responses to Questions

**Comments to the Author**

1. Is the manuscript technically sound, and do the data support the conclusions?

Reviewer #1: Yes

Reviewer #2: Yes

2. Has the statistical analysis been performed appropriately and rigorously? 

Reviewer #1: Yes

Reviewer #2: Yes

3. Have the authors made all data underlying the findings in their manuscript fully available?

Reviewer #1: Yes

Reviewer #2: Yes

4. Is the manuscript presented in an intelligible fashion and written in standard English?

Reviewer #1: Yes

Reviewer #2: No

5. Review Comments to the Author

Reviewer #1: Jocelyn et al. investigated the role of Colchicine pre-treatment on oleic acid-induced lung injury in rats. They found that Colchicine reduced ALI and respiratory failure in experimental ARDS concerning reduced lung neutrophil recruitment and reduced circulating leukocyte activation. They also found that increased lung NETosis was also reduced by Colchicine administration therapy following oleic acid. This is an exciting study showing an essential role of neutrophil in the lung following oleic acid-induced injury. It would strengthen the studies and support the conclusions if the authors could employ alveolar macrophage-specific experiments. Overall, this paper attempts to answer some interesting questions, but does not provide new mechanistic insights into how Colchicine modulates host immune response. The manuscript is well written.

Reviewer #2: The manuscript entitled “ Colchicine reduces lung injury in experimental acute respiratory distress syndrome” by Dupuis et al. explained Colchicine's role in reducing the acute respiratory distress syndrome in an animal model. They induced the ARDS by oleic acid, and they use three groups of rats and each group containing 8 rats. They examined the different biological markers in the sham, OA control and OA + colchicine groups. Finally, they found the protective effect of Colchicine in reducing the ARDS. I think the manuscript has substantial merit for publication after some minor revision.

1. The authors wrote, ‘ 2.5% isoflurance. Actually, it should be ‘isoflurane’.

2. Some grammatical errors are found throughout the manuscript that should be corrected.

6. PLOS authors have the option to publish the peer review history of their article (what does this mean?). If published, this will include your full peer review and any attached files.

Reviewer #1: No

Reviewer #2: **Yes: **Mohammad Safiqul Islam

---

## [Author Response · Author response to Decision Letter 0]

29 Oct 2020

Response to reviewers.

We would like to thank again both reviewers and academic editor for their interest to review our work and for providing thoughtful insight. We have performed modifications according to the comments received. We also decided to remove the statement that 6000 subjects will be recruited in the COLCORONA study as this number may change following interim analysis.

Academic editor: Thank you for submitting your manuscript to PLOS ONE. After careful consideration, we feel that it has merit but does not fully meet PLOS ONE’s publication criteria as it currently stands. Therefore, we invite you to submit a revised version of the manuscript that addresses the points raised during the review process.

As you can see from the reviewers comments below, the required changes are relatively minor. Upon reading the manuscript myself, I also noted a couple of minor issues, which are listed below as well. 

As the academic editor, I have two additional comments to those provided by the two reviewers.

1) Please include a short discussion regarding the limitation of utilizing pretreatment for ARDS.

2) Please include scale bars in the histological images.

Reply: We thank the academic editor for his review and comments on our paper.

1) We added the following paragraph to the “Limitations and perspective” section: “Our results must be interpreted in the specific context of pre-treatment with colchicine to prevent ARDS and cannot be extrapolated to efficacy for the treatment of florid ongoing ARDS.” 

2) Fig 2 has been modified to include scale bars as requested.

Reviewer #1: Jocelyn et al. investigated the role of Colchicine pre-treatment on oleic acid-induced lung injury in rats. They found that Colchicine reduced ALI and respiratory failure in experimental ARDS concerning reduced lung neutrophil recruitment and reduced circulating leukocyte activation. They also found that increased lung NETosis was also reduced by Colchicine administration therapy following oleic acid. This is an exciting study showing an essential role of neutrophil in the lung following oleic acid-induced injury. It would strengthen the studies and support the conclusions if the authors could employ alveolar macrophage-specific experiments. Overall, this paper attempts to answer some interesting questions, but does not provide new mechanistic insights into how Colchicine modulates host immune response. The manuscript is well written.

Reply: We would like to thank the reviewer for this evaluation and interesting suggestion. We agree that our study would be further improved by additional mechanistic insight on the effect of colchicine on the innate immune response and that studying alveolar macrophages would be an interesting way to do so. We now recognize this in the: “Limitations and perspective” section as follows: 

“Although colchicine reduced lung neutrophils recruitment and activation, other potential mechanisms of action of colchicine on the innate immune host response, especially on alveolar macrophages, were not evaluated. These will require further studies.”

Reviewer #2: The manuscript entitled “Colchicine reduces lung injury in experimental acute respiratory distress syndrome” by Dupuis et al. explained Colchicine's role in reducing the acute respiratory distress syndrome in an animal model. They induced the ARDS by oleic acid, and they use three groups of rats and each group containing 8 rats. They examined the different biological markers in the sham, OA control and OA + colchicine groups. Finally, they found the protective effect of Colchicine in reducing the ARDS. I think the manuscript has substantial merit for publication after some minor revision.

1. The authors wrote, ‘ 2.5% isoflurance. Actually, it should be ‘isoflurane’.

2. Some grammatical errors are found throughout the manuscript that should be corrected.

Reply: We thank the reviewer for his revision and suggestion to improve the paper. 

1) The typo has been corrected.

2) We reviewed the manuscript in order to correct grammatical errors and remove repetitions. We hope these modifications have improved the readability of our paper.

---

## [Editor Report · Decision Letter 1]

2 Nov 2020

Colchicine reduces lung injury in experimental acute respiratory distress syndrome

PONE-D-20-28950R1

Dear Dr. Dupuis,

We’re pleased to inform you that your manuscript has been judged scientifically suitable for publication and will be formally accepted for publication once it meets all outstanding technical requirements.

Kind regards,

Ruud AW Veldhuizen

Academic Editor

PLOS ONE

---

## [Editor Report · Acceptance letter]

4 Nov 2020

PONE-D-20-28950R1 

Colchicine reduces lung injury in experimental acute respiratory distress syndrome 

Dear Dr. Dupuis:

I'm pleased to inform you that your manuscript has been deemed suitable for publication in PLOS ONE. Congratulations! Your manuscript is now with our production department. 

Kind regards, 

on behalf of

Dr. Ruud AW Veldhuizen 

Academic Editor

PLOS ONE